# A Stereometry of Non-Memory: Mapping a Lost Past in W.G. Sebald's *Austerlitz*

Michael Holden

Department of Geography, Royal Holloway, University of London, Egham TW20 0EX, UK;
michaeljoseph.holden@rhul.ac.uk

**Abstract:** This article presents a consideration of W.G. Sebald's 2001 work *Austerlitz*—his final novel—according to a variety of spatial and cartographic concepts, including 'fluid cartography,' and the notion of countermapping. Particularly, the article will explore the eponymous protagonist's sense that 'time [does] not exist at all, only various spaces interlocking according to the rules of a higher form of stereometry,' and will demonstrate how this subjective experience of time is a consequence of the absence of memory experienced by the protagonist in relation to his origins as a Kindertransport survivor of the Holocaust. Similarly, the article will explore how spaces—particularly buildings—and material artefacts come to act as an (insufficient) surrogate for memory within the text. All of the above will be framed according to a reading of the fundamental spatiality of Sebald's works, and particularly their map-like quality.

**Keywords:** Holocaust; space; memory; literary representation; fluid cartography; Sebald

## 1. Introduction

'Has there ever been,' asks John Banville, 'a more devastating and yet wholly undemonstrative account of the mid-20th century European horrors as [*sic*] *Austerlitz* Sebald's (2002a) final novel; his masterpiece, and one of the supreme works of art of our time?' (Banville 2003, ¶2). It would seem that its 'undemonstrative' quality is here perceived as a positive attribute, and is crucial to Banville's enthusiastic reception of the novel. Whilst it is not hinted at in this description, however, I suggest that this quality of being 'undemonstrative' originates from the void that is at the novel's core—namely, a void of repressed memory, from which the protagonist struggles throughout to emerge. It is owing to this mnemonic black hole that the text relies on 'allegorical indirection' (Theisen 2006, p. 563) and a 'spherical system of association and encyclopedic links'—these are the very characteristics that the scholar Bianca Theisen identifies as being central to the author's earlier work, *The Rings of Saturn* (Theisen 2006, p. 569). *Austerlitz* differs somewhat, in that it explores the suppression of its protagonist's natural recollections and instead employs 'association and encyclopedic links' in the role of a kind of surrogate memory.

I have elsewhere described the 'map-like' quality of Sebald's work—particularly in *The Rings of Saturn*—and the ways in which the author's map-like constructions are inseparable from the mnemonic concerns of his texts. Crucial to this aspect of his representations of space is an array of real-world markers of place, extended perambulations on the part of protagonists and other characters, and a proliferation of overhead and bird's-eye views (to give but a few examples), and it is within such spaces (and the mental wanderings they inspire) that memory and history come to bear upon the narrative (see Holden 2021). Particularly, I have demonstrated the complementary fluidities of the memories and cartographies expressed within Sebald's texts, particularly with reference to the notion of 'fluid cartography,' a critical understanding of mapping introduced by Isabel Capeloa Gil and João Ferreira Duarte. They describe fluid cartographies as those that '[aim] at understanding instead of controlling,' and which

> address the fluid disengagement of the modern world, the diasporic displacements and the complex changes that mark the transitive and transitional reality of modernity . . . a fluid cartography, moreover, traces connections in contact zones and perceives the limits that mark the territory not as borders but rather . . . borderlands . . . A fluid cartography, then, perceives the territory as an emerging surface where charting is equated with inscribing and translating, where different identities, times and locations come together. (Gil and Duarte 2011, p. 3)

Given Sebald's idiosyncratic fondness for drawing connections between disparate histories that diverge across time and space, the utility of this concept is clear. In light of the intimate intermingling of space and memory in his writing, moreover, it follows that memory, too, can be considered a fluid entity in his works.

*Austerlitz* is no different, but I will argue here that association and 'allegorical indirection' also figure as a kind of surrogate for individual memory where genuine recollection is inaccessible owing to its repression. The text is predominantly concerned with recounting the search of the eponymous protagonist for his lost past, particularly in the form of his family history and his own memories. The life of Jacques Austerlitz is explained to the narrator at a series of (apparently) chance meetings, in moments at which the paths of the two men intersect at various locations across Europe. As such, this article will be concerned with demonstrating the more abstracted, allegorical character that colours the 'fluid cartography' within the pages of *Austerlitz* when compared with earlier works such as *The Rings of Saturn*.

*Austerlitz* does not only recount a (more-or-less) linear journey across readily-mappable spaces, spinning off centrifugally from its waypoints in meditations on cultural memory and history in the manner of *The Rings of Saturn* (Holden 2021). While this technique remains fundamental here too, in *Austerlitz* the author also interweaves additional, complex temporal layers that entangle multiple strands of the narrative itself. Naomi Stead notes that:

> . . . dream-like conversations . . . composed of long monologues from Austerlitz retold by a self-effacing and unnamed narrator . . . take place in a series of carefully evoked real locations: a bar at Liverpool Street Station in London, the Le Havane café at number 70 on the Boulevard Auguste Blanqui in Paris, where they meet on three occasions, and various places in the city of Antwerp. (Stead 2015, p. 41)

So, the fragmentary story of Austerlitz is recounted to the reader via the intermediary listening (and writing) figure of the narrator—indeed, Austerlitz's story comes to form the bulk of the text. By contrast, in earlier works it is often the journey undertaken by the narrator himself that is recounted; here, that journey—whilst still present—becomes subsumed within Austerlitz's narrative.

Crucially for my purposes, toward the novel's end the eponymous protagonist himself makes the following observation about his experience of space and time: 'I feel more and more as if time did not exist at all, only various spaces interlocking according to the rules of a higher form of stereometry, between which the living and the dead can move back and forth as they like' (p. 261). Stereometry refers to the volumetric measurement of three-dimensional Euclidean space—or, in other words, the measurement of (representations of) physical space. I suggest that this subjective experience of spatio-temporality represents something closely akin to the narrative structure of the novel itself, particularly in the notion of 'various spaces interlocking'—we might recall here Foucault's notion of the heterotopia as comprising 'slices of time' (Foucault 1986, p. 26). In a similar vein, I will also outline how the text appears, in some respects, *circular* in character. While this is also a characteristic of earlier works such as *The Rings of Saturn*, this cyclical nature pervades the structure of *Austerlitz*, and is pronounced to a greater degree. As we will see, images and spectral associations recur throughout.

Both cyclicality and a patchwork approach to narrative temporality are apposite to—or perhaps precede—the specific character of the fluid cartography on display in *Austerlitz*. While typical Sebaldian features such as those already mentioned—bird's-eye views and

heightened vantage points, mimetic representation of space, a multitude of specific place-markers, wanderings, transhistorical musings, and so on (Holden 2021)—are once again in abundance, the relationship of such features to the narrative of the text becomes profoundly altered. The uncertainty at the novel's core—predicated on the disintegrated childhood memories of its protagonist and the concordant, traumatic fragmentation of his sense of self—serve to destabilise the text and its fluid cartography. Thus, due to its substantial length, dense structural complexity, patchwork temporality, and its copious, interwoven cultural and historical information, *Austerlitz* itself feels like a monumental construction that is barely capable of sustaining its own weight. Stead suggests that:

> [the] ambiguity of Sebald's work has often been noted—his blurring of fact and fiction, his elusive use of uncaptioned photographs, and so on . . . [but] amidst all Sebald's ambiguities, it is architecture in its massive and solid materiality which acts as an anchor tying the fictional narrative to the material world. (Stead 2015, p. 41)

The reading I will offer here, however, differs slightly. Rather than truly functioning as an 'anchor,' I contend that such 'massive and solid materiality' instead represents merely an attempt by the protagonist to 'anchor' his own narrative in the manner described by Stead, as a means of overcoming his unstable, rootless identity. This attempt is ultimately unsuccessful. Sebald's final novel is particularly concerned with architecture—or rather, '*Austerlitz*'s overriding aversion to architectural giganticism'—as well as objects, monuments, and museums (Stead 2015, p. 42). Consequently, we find a collection of reproduced maps, diagrams, and textual artefacts included within the novel as supplements to the map-like narrative construction. In what follows, I will demonstrate how this concern with material memory is expressed in the structure of the novel itself. Indeed, this article will focus not only upon the spaces represented within the text, but also the space of the novel-as-text, and the borderland between the physical page and the story written upon it; in other words, I will be concerned with both the spaces within the novel and the notion of the novel as an exercise in spatially structuring a narrative. Apposite to the repressed Holocaust memory at its core, we might read *Austerlitz* as a narrative that remains unmoored and fragmented, in contrast with a work such as *The Rings of Saturn* which, despite its intricacy, is fundamentally grounded in the physical act of walking the Suffolk coastlands.

Latent within the characteristics listed above is a reaction to linear or monumental conceptions of history and memory. With respect to the specific material concerns of the text, Stead proffers a threefold conception of the significance of architecture: '. . . as premonition of disaster, as monument to barbarism, and as instrument of oppression' (Stead 2015, p. 42). In analysing the narrator's description of Brussels's imposing *Palais de Justice*, Stead observes that within this vast structure 'the fine symbolism of justice and civic life is transformed into . . . a blind labyrinth signifying only state power and its inscrutable bureaucratic processes'; she continues: 'In this case . . . Austerlitz identifies that the rational procedures of Enlightenment thought hold within them the very seed of inhuman domination' (Stead 2015, p. 43). In a related sense, Laura García-Moreno observes that, '[from] Austerlitz's perspective, grid-like patterns are indicative of a compulsion to control and regulation' that both character and author alike 'view with profound suspicion' (García-Moreno 2013, p. 367).

Instead, in *Austerlitz*, 'non–grid-like connections between people, places, times, and objects proliferate' and paths 'constantly [cross] over and [overlap]' (García-Moreno 2013, p. 367). This characterisation gestures toward the particular narrative method of *Austerlitz* that I began to outline above. Broadly, the work represents a continuation of the notion—expressed by Theisen—that Sebald's writing 'rediscovers allegorical indirection as perhaps the more appropriate . . . approach to the labyrinthine truths of reality,' while seeking to temper the 'narrative of progress embraced by those who believed they left the age of darkness for enlightened analysis' (Theisen 2006, p. 563). In *Austerlitz*, we see an exploration of 'the fragility of human memory in the face of the crushing forces of history' (Stead 2015, p. 42). Any 'excavation of the past' is necessarily stymied by the fact that

the protagonist is mired in absence and loss (García-Moreno 2013, p. 365). As such, a fluid cartography in the manner described by Gil and Duarte is, perhaps, apposite to a more relational rendering of history or memory. I will demonstrate how Sebald's *Austerlitz* reflects the protagonist's feeling of 'various spaces interlocking according to the rules of a higher form of stereometry,' whilst introducing associative connection as a substitute for stifled, hidden memories—we may regard the novel, accordingly, as a kind of stereometry of non-memory (p. 261).

In essence, all of the above emerges as a response to the disjunction and trauma occasioned by the Holocaust. Despite the centrality of the Nazi genocide to the novel's narrative progression, the topic is treated here—as within all of Sebald's works—quietly and gently; the author evokes this history with a 'murmur,' rather than a shout, as Banville has it (Banville 2003, ¶2). Indeed, the Holocaust itself must be unearthed by the protagonist as a dimension of his own past, given that he escaped its horrors on a Kindertransport train (hence the text's mnemonic black hole). Broadly, I will argue that *Austerlitz* takes up the narrative method of earlier works such as *The Rings of Saturn*—centrifugal meditations upon a 'natural history of destruction'—and amplifies and complicates it. This is a direct consequence of its subject matter, and particularly due to the fact that absences and voids figure heavily within the novel in a manner in which they (usually) do not in the author's earlier works. The (non-)memories of Jacques Austerlitz (and the history they represent) are, in the words of Henri Raczymow, '*la mémoire trouée*'—a memory 'shot through with holes' (Raczymow 1994). As such, the map-like text and the journeys it describes themselves become similarly perforated.

## 2. Slices of Time: A Stereometry of (Non-)Memory

> When I finally went over to Austerlitz with a question about his obvious interest in the waiting room, he was not at all surprised by my direct approach but answered me at once, without the slightest hesitation, as I have variously found since that solitary travellers, who so often pass days on end in uninterrupted silence, are glad to be spoken to. (p. 7)

The above passage describes the moment of the narrator's first encounter with the eponymic protagonist, and it is around these two 'solitary travellers' that the narrative revolves. It marks the first of the novel's 'dream-like conversations'—its 'long monologues from Austerlitz ... retold by a self-effacing ... unnamed narrator,' which take place in 'carefully evoked real locations'—here the waiting room of the railway station in Antwerp (Stead 2015, p. 41). In the chance meetings between narrator and character that make up the text we see an example of heterotopic 'slices of time,' which here make up the building blocks of what I am choosing to describe, echoing Sebald's protagonist, as a 'stereometry of (non-)memory.' These coincidental meetings—situated as they are in a multitude of specific, real-world locations—provide the clearest example of the 'interlocking' spaces described by Austerlitz. Narratively, they appear to neatly slot in alongside one another regardless of how much time has elapsed between them (a fact remarked upon by the narrator several times), and they become the platform upon which the novel's labyrinthine, 'dream-like conversations' on topics of memory and history unfold (p. 41). Moreover, with respect to the notion of fluid cartography, the interlocking spaces within which the conversations take place stage the blurring of a number of boundaries—between past and present, between different places evoked by the interlocutors, and even between different characters, particularly the narrator and Austerlitz themselves. Indeed, in this regard, Stead describes the narrator as a 'self-effacing figure' who 'shares many characteristics with the author' (thereby adding an additional layer of boundary-blurring, here between the text and the real world), who 'reveals very little of himself,' and instead '[takes] the role of bearing witness' (Stead 2015, p. 42). The boundary between the two figures (and perhaps even with the author, too), becomes somewhat blurred within the text. In Gil and Duarte's terms, rather than a definitive 'border' between the characters there exists, rather, a sort of 'borderland,' in which the narrator's self-effacing nature allows him to become,

in some regards, a conduit for Austerlitz (or perhaps a conduit between character and author).[1] This all occurs alongside the tracing of tales which map out a vast amount of real geographical information.

This boundary-blurring is perhaps best exemplified in the monumental sentences with which *Austerlitz* is constructed. Often, Sebald presents a sort of 'said *x*, said *y*, said *z*' syntactic structure, with the narratives of a multitude of characters nested within one another like a set of Matryoshka dolls. In every case, reported speech is filtered through the narrator in the first instance, but usually also through Austerlitz himself; the reader often receives retold narratives at a minimum of a double remove. We see this in the following example, which derives from Austerlitz's retelling of his return to Prague in search of some element of his history, at which point he is reunited with Vera, his childhood nanny:

> *I believe, Vera told me, said Austerlitz*, that even the last remaining German sceptics were overcome by a kind of euphoria, such as one feels at high altitude, in these years when victory followed upon victory, while we, the oppressed, lived below sea level, as it were. (pp. 248–49, emphasis added)

García-Moreno suggests that '[t]he frequently overlapping temporalities and different interpretive frames' that are characteristic of *Austerlitz*'s sentences have the effect (in tandem with the irruption of uncaptioned images into the text) of 'slow[ing] down the reading process and disrupt[ing] the flow characteristic of narrative, creating the sense of a complex, expanded, almost *spatialized* duration' (García-Moreno 2013, p. 361, emphasis added). This is true, but I would add that such sentence construction—particularly in tandem with the lack of firmly demarcating punctuation marks between speakers—also entails the boundary-blurring effect described above. At times, it is easy to confuse precisely who is recounting a particular incident and, particularly, to remember that the entire narrative is being relayed to the reader not by Austerlitz, but by the narrator.

This is in part a consequence of the 'self-effacing' nature of this figure—the story, after all, largely belongs to the protagonist. At moments such as the above, however, the nature of the novel's dialogue serves to blur the boundaries between various characters and, particularly, the times and spaces within which they find themselves. Take, for instance, Vera's recollection of an exclamation made by Agáta (Austerlitz's mother), on the topic of a Nazi proclamation: 'The Jew concerned in the transaction! Agáta had cried, adding: Really, the way these people write! It's enough to make your head swim' (p. 249). Here, we are hearing Agáta's words at a threefold remove (Vera, Austerlitz, narrator), yet her exclamations ring out clearly as if they are an example of direct, rather than reported, speech. Similarly, as Vera recalls the distressing moment of Agáta's deportation, we see a similar effect:

> Agáta soon asked me to leave her. When we parted she embraced me and said: Stromovka Park is over there, would you walk there for me sometimes? I have loved that beautiful place so much. If you look into the dark water of the pools, perhaps one of these days you will see my face. Well, said Vera, so then I went home. It took me over two hours to walk back to the Šporkova. I tried to think where Agáta might be now, whether she was still waiting at the entrance or was already inside the Trade Fair precinct. (p. 253)

She goes on to describe how she later learned, from a survivor, what conditions were like in the Trade Fair precinct, providing us with an account that is at yet another, different remove from the reader and narrator. From this account, Vera provides Austerlitz with a minutely detailed description of the horrors of deportation from Prague. In all of the above, the boundaries between times, spaces, and individuals become blurred; we are variously (and simultaneously) privy, for instance, to the space within which Austerlitz and the narrator are conversing (p. 234; here, unusually, it is Austerlitz's own house), Vera's living room, and the spaces and places of the wartime Prague of Vera's young adulthood. 'Stromovka Park is *over there*,' for instance, suggests a spatio-temporal and subjective immediacy that would be lost in a construction such as 'she gestured towards Stromovka Park,' or similar.

Likewise, Vera notes that she 'tried to think where Agáta might be *now*,' thereby lending the recollection a similar sense of imminence and further confusing temporal boundaries. Austerlitz's sense of 'spaces interlocking according to . . . a higher form of stereometry' is pronounced here, and in this instance may be understood to constitute the narrative itself, and its labyrinthine construction (p. 261).

This perception of blurred boundaries and interlocking spaces is notably heightened by the relationship between Austerlitz and the narrator. The narrator observes that, strangely, in every encounter with Austerlitz after the first, 'we simply went on with our conversation, wasting no time in commenting on the improbability of our meeting again in a place like this, which no sensible person would have sought out' (p. 37). This unusual relationship—which even appears to negate their separation for a substantial period of years—renders time and space (seemingly) irrelevant; the two characters appear to move freely between Austerlitz's 'interlocking' slices of time and space, rather than being bound to more conventional temporal progression. Moreover, the spaces in which the characters find themselves seem devoid of symbolic significance bar their relationship to time (or rather, reluctance to accept its onward march). These are typically spaces in which time seems to stand still. They meet in waiting rooms, gloomy cafés, Austerlitz's own peculiarly austere house, and (somewhat pointedly) the Greenwich Royal Observatory, the Ur-site for our contemporary understanding of time and the artificially fixed baseline to which other time zones relate.

A similar effect is seen when Austerlitz first re-encounters Vera in Prague. After requiring an English-speaking member of staff on the previous day to assist him in his struggle against the bureaucracy of the State Archives,[2] Austerlitz finds that, inexplicably, after some time with Vera he is able to speak Czech—the language of his pre-*Kindertransport* childhood—once more:

> In the middle of her account Vera herself, quite involuntarily, had changed from one language [French] to the other [Czech], and I, who had not for a moment thought that Czech could mean anything to me, not at the airport or in the state archives . . . now understood almost everything Vera said . . . so that all I wanted to do was close my eyes and listen for ever to her polysyllabic flood of words. (p. 219)

Austerlitz's experience of time and space as an interlocking stereometry is again pronounced here—it is as if he has stepped directly into his early life, seemingly untouched by the traumatic displacements of the intervening years (as per Gil and Duarte 2011, p. 3). As Austerlitz notes: 'I found myself back among the scenes of my early childhood, every trace of which had been expunged from my memory' (p. 212) yet, despite this mnemonic void, he finds (in Proustian fashion) that: 'when I felt the uneven paving of the Šporkova underfoot . . . it was as if I had already been this way before and memories were revealing themselves to me not by means of any mental effort but through my senses' (p. 212). It is significant that spaces of home (the street Šporkova) and domestic spaces (Vera's apartment) spark some form of recollection in the memory-vacuum, rather than the official, bureaucratic space of the State Archives, which he finds immensely oppressive (p. 208).

In all of the above, Gil and Duarte's notion of fluid cartography as a kind of melding of 'different identities, times and locations,' is clear (Gil and Duarte 2011, p. 3). Indeed, such flexible spatial representation is necessary to accommodate mnemonic and historical meditations such as these. Temporally and geographically distant spaces, for instance, appear to interlock with one another to create a hybrid space from past(s) and present, here and multiple over-theres. Nonetheless, mimetic specificity of place is maintained throughout. Characters, meanwhile, seem to traverse these borderlands quite easily, and at times appear to become somewhat indistinct as separate entities, as the labyrinthine quality of Sebald's prose weaves the voices of multiple interlocutors into an intricate and complex whole. Likewise, the narrative method employed by Sebald might be viewed as a rebuttal to the notion of a purely linear retelling of history, such as that often seen in actual historical accounts, or in the narratives of historical novels. It can be understood, in essence, as charting an alternative path for the presentation of historical narratives (albeit here in

fictional form). Tessa Morris-Suzuki, in *The Past Within Us*—her study of the 'historical truthfulness' of cultural works that seek to represent history—observes of traditional (i.e., typically, nineteenth-century) historical novels:

> The historical novel . . . creates a new form of empathic link between past and present, between the lives of readers and an imagined image of the society of the past. But at the same time it also frames that society spatially, most often in terms of the nation state. In this way, it has been one of the chief vehicles through which the peoples of modern times were encouraged to imagine the past in national terms. Both modern history writing and the modern novel itself are inextricably linked to processes of nation-building. (Morris-Suzuki 2005, p. 49)

This national(istic) spatial framing is so pronounced in such works, Morris-Suzuki suggests, that '[n]ational maps play an important part in historical fiction, often appearing in the opening pages to fix the boundaries of the novel's action' (Morris-Suzuki 2005, p. 51). Sebald's works—whilst profoundly concerned with representing specific places—present historical narratives in a vastly different way, offering instead a patchwork of historical moments and images that tend to span a wide array of times and spaces. Even within the relatively bounded first-hand experiences of his protagonist in *Austerlitz*, Sebald guides his reader across multiple national boundaries, a variety of times and spaces, and even multiple languages, all the while retaining, through narrative technique, a sense of the seamless interlocking of a multitude of Foucauldian 'slices of time,' or Austerlitz's own 'stereometry.' What Morris-Suzuki describes as the 'cartographical *mise en scène*' of the historical novel becomes, in Sebald's hands, a miscellany of discreet, varied, yet connected times and spaces (Morris-Suzuki 2005, p. 51).

### 3. Memory by Association

Equally crucial to the 'stereometry of memory' upon which the narrative of Austerlitz is constructed is an array of cyclically recurring, uncanny, visual and spatial associations. Early in the text we are told how Austerlitz (being an architectural historian) wishes to create a gigantic synthesised history of the 'architectural style of the capitalist era,' an endeavour which he describes in the following terms:

> Why he had embarked on such a wide field, said Austerlitz, he did not know; very likely he had been poorly advised when he first began his research work. But then again, it was also true that he was obeying an impulse which he himself, to this day, did not really understand, but which was somehow linked to his early fascination with the idea of a network such as that of the entire railway system. (pp. 44–45)

This notion of a vast, intricate network is evident in much of the imagery of *Austerlitz*, whereby particular pictures, shapes, and ideas appear to re-emerge at multiple moments throughout, thereby linking discrete moments of the text at which very different issues are under discussion. Often, they seem to gesture in a ghostly manner toward the protagonist's profound repression of his past, suggesting a fluid interrelation between the *then* of the past and the *now* of the narrative. Such cyclical associations thereby strengthen the sense of the text as being composed of a kind of 'stereometry' of memory. In this sense, much like the narrative techniques discussed above, we can view these repeated images as signifiers of a permeable 'borderland' (to use Gil and Duarte's terms) between different (temporal and spatial) moments within the text.

A particularly pointed example can be found in the figure of Lake Vyrnwy, a reservoir near the home in Wales into which Austerlitz was adopted after arriving in Britain. Early in the text, we are told that the young Austerlitz, when lying in bed at night, 'often felt as if [he] too had been submerged in that dark water,' like the town which made way for the reservoir, and that 'like the poor souls of Vyrnwy must keep [his] eyes wide open to catch a faint glimmer of light far above [him]' (p. 74). This is one of many references to bodies of water within the text. Frozen rivers and lakes are particularly pronounced, such as an

early description of Lucas van Valckenborch's sixteenth-century painting of 'the frozen [river] Schelde' in Antwerp (p. 15); later, we are offered a description of the frozen 'Lake Bala,' a neighbour to Lake Vyrnwy (p. 89). A number of other lakes and rivers—frozen or otherwise—are referred to throughout the text, but it is not until much later that the significance of this recurring image is revealed. While travelling back from Prague by rail, toward the Hook of Holland (recreating his hitherto-forgotten childhood evacuation on a *Kindertransport*), Austerlitz passes through Germany:

> And then, Austerlitz continued, somewhere beyond Frankfurt, when I entered the Rhine valley for the second time in my life, the sight of the Mäuserturm in the part of the river known as the Binger Loch revealed, with absolute certainty, why the tower in Lake Vyrnwy had always seemed to me so uncanny. (p. 317)

An uncaptioned photograph inserted into the midst of this paragraph signals to the reader (through implication) that this sense of the uncanny derives from the visual similarity of the two vistas—this is founded upon the presence of an ornate tower in the middle of a large, narrow body of water that is surrounded by hills. Austerlitz goes on to describe—in decidedly Romantic terms—the sublimity of this encounter. As he passes by, the sun breaks through clouds and fills the valley, and he finds himself strongly (and unaccountably) moved to interpret the landscape as 'prehistoric and unexplored' (p. 318).[3] The apparently revelatory nature of this moment begins to account for the proliferation of images of lakes. The young Austerlitz's sense of being trapped underwater, for instance, might now be read as a manifestation of repressed memories; the repeated images and discussions of frozen lakes, meanwhile, might now be interpreted as a symbol of frozen memory, suddenly thawed by the breaking of sunlight on the Mäuserturm, the tower in the Binger Loch.[4] These interpretations are, of course, speculative—such decisive conclusions are not made explicit in the text. What is clear, though, is that the repetition of such images suggests a larger question of symbolic connection (in my view, along the lines of memory, frozen and submerged) and, particularly, a strong sense of the interconnectedness of the places and moments described. Lucas van Valckenborch's sixteenth-century painting, post-war Wales, and Germany in the present-tense of the narrative (to name but three examples) all seem to map out a symbolic network of connection that runs throughout the text, and thereby recall the 'higher form of stereometry' described by Austerlitz. The array of such images seems to marry the individual to the flow of history by enmeshing the protagonist's personal recollections and experiences into a much broader network of related images, many of which are associated with times and places far removed from the protagonist's journey. Via their wide-ranging discussions, Sebald's narrator and protagonist map out eccentric networks of association that forego any sense of spatial or temporal linearity.

Such networks of association recur throughout the text but, owing to constraints of space, I will not devote much more time to discussing them here, barring one final, brief example. García-Moreno notes of Sebald's works that,

> [c]onnections between time and space, as well as between past and present, cannot be thought of in terms of continuity or unity. A traumatic past has disabled the present and irrevocably derailed its connection to the past, which, in turn, marks the present to the core without the subject being fully aware of how or why. (García-Moreno 2013, pp. 361–62)

Such a sense of connection that ultimately rests on disjunction and a lack of the 'how' or 'why' accounts for the uncanny nature of Austerlitz's encounter with the Mäuserturm, and can also account for the 'stereometry' of interlocking spaces that is experienced by the protagonist (and is mapped out by the narrative). A telling example is to be found in recurring discussions of fortifications and an array of other star-shaped structures that reappear throughout the text. These moments represent a chain of connection that ultimately leads Austerlitz to Terezín, which is where he suspects that his mother perished. Throughout the novel, we trace this chain of connection, which runs from the 'tendency towards paranoid elaboration' (p. 19) attributed to the star-shaped German-French border town of Saarlouis,

through the crab-like fortress of Breendonk, a feverish dream of being trapped 'at the innermost heart of a star-shaped fortress' when Austerlitz first unearths his true origins (p. 196), to the star-shaped walls of Terezín itself (pp. 280, 328–29), interspersed throughout with a number of other star-shaped objects and diagrams. Again, it is only upon the revelation of Austerlitz's past and his attempt to discover traces of his family that we can see with clarity the poignant significance of this recurring image. Each example of a fortress (or of more general star-shapes) represents one of the interlocking spaces that Austerlitz describes as composing the 'higher form of stereometry' that he feels governs his life (p. 261). It is as if we have, as readers, gradually stepped through a variety of these spaces alongside the protagonist, as we follow him in edging closer to his origins throughout the novel.

Once again—in Gil and Duarte's terms—a set of permeable borderlands are created between distinct times and spaces. Indeed, as Austerlitz himself notes of his sense of a 'higher form of stereometry,' the 'interlocking' spaces of which it is composed act as a conduit 'between which the living and the dead can move back and forth as they like' (p. 261). It is as if these spaces and imagic associations accrete into a palimpsest, thereby weaving a series of oblique collages of repressed memory throughout the novel. Gil and Duarte's sense of fluid cartography as a means of representing, or voicing, the experience of 'diasporic displacements' is likewise relevant; much of the (apparent) significance of these cyclically recurring associations derives from a past that was lost to the protagonist when he was removed from the world of his childhood on a *Kindertransport* and transplanted, alone, into the unfamiliar environment of Wales. The novel can, as such, be read as a poignant example of just such displacement. Likewise, the text's recurring images may also be viewed as offering an alternative to more conventional, linear presentations of historical narratives. Here, rather than offering a temporally linear account, Austerlitz's history is alluded to in a range of symbolic images, which coalesce into the temporally and geographically fluid space of the text itself. The significance of these image-chains is only revealed (obliquely) when the past of the protagonist becomes clear. This past, then, is linked by association to an array of discrete times and spaces which play out within the 'interlocking' spaces of the text's 'higher form of stereometry'.

Once again, we might recall García-Moreno's assertion of Austerlitz's (and Sebald's) aversion to grid-like patterns, which are viewed as 'indicative of a compulsion to control and regulation' (García-Moreno 2013, p. 367). Rather, it is ultimately the 'non-grid-like connections between people, places, times, and objects' that form the interlocking spaces of Austerlitz's sense of a guiding 'stereometry' (p. 367). His visit to the Terezín Ghetto Museum, for instance, inspires relative incomprehension. Here, he '[studies] ... maps of the Greater German Reich,' 'traces [their] railway lines,' sees evidence of the Nazi 'mania for order and purity,' and for 'obsessive organizational zeal,' and observes 'plots of land meticulously parcelled out' in diagrammatic form, as well as their 'balance sheets, registers of the dead, lists of every imaginable kind' (García-Moreno 2013, pp. 278–79). Despite this vast wealth of meticulous information—the 'incontrovertible proof' which opens up to him 'the history of persecution which [his] avoidance system had kept from [him] for so long' (pp. 278–79)—he notes that:

> I understood it all now, yet I did not understand it, for every detail that was revealed to me as I went through the museum from room to room and back again, ignorant as I feared I had been through my own fault, far exceeded my comprehension. (p. 279)

This documentary evidence, it seems, serves only to push the protagonist further from this aspect of his own history; it offers only a kind of horrified bewilderment. Such a reaction is of course—in some regards—a wholly appropriate reaction to the crime of the Holocaust, but it is scarcely helpful for a man who is seeking some understanding of his own absent history.

By contrast, Austerlitz also relates to the narrator his experience of the 'A N T I K O S B A Z A R [*sic*],' an antique shop in Terezín (Figure 1) (p. 273). He first introduces this peculiar shop in the context of a transition between dream and waking memory, whereby he recalls how, one morning, he dreamt that he was looking into a Terezín barrack; while

trying to cling onto this 'powdery grey dream image,' he finds that it becomes 'overlaid by the memory, surfacing in [his] mind at the same time, of the shining glass in the display windows of the A N T I K O S B A Z A R' (pp. 272–73). The window display—consisting of 'four still lifes obviously composed entirely at random,' made up of discarded bric-a-brac— 'exerted such a power of attraction' over the protagonist that 'it was a long time before [he] could tear himself away' (p. 274). Standing in front of this strange display, Austerlitz seems to embark upon a profound search for meaning, in stark contrast to his encounter with the Ghetto Museum, which follows shortly afterwards: 'What was the meaning of the festive white lace tablecloth hanging over the back of the ottoman, and the armchair with its worn brocade cover?'; 'What secret lay behind the three brass mortars of different sizes, which had about them the suggestion of an oracular utterance . . . ?' (p. 275).[5] These are but two in a sequence of seemingly endless questions, all of which lead to the recollection of the Czech word for squirrel—'*veverka*'—which is prompted by the sight of a stuffed *veverka* in the display. While revealing nothing in the way of concrete information, this curious collection of miscellaneous artefacts inspires some sense of connection, at least, between Austerlitz and his lost past, signalled here through the excavation of a forgotten word. We see a similar effect at play in the moment—often commented upon by scholars—at which Austerlitz attempts to catch a glimpse of his mother in a film of Terezín produced by the Nazis after the infamously stage-managed visit of the Red Cross (p. 342). Here, incomprehension and bewilderment reign until Austerlitz decides to alter the temporality of the video by slowing it down, so that 'the men and women employed in the workshops now looked as if they were toiling in their sleep' (p. 345) and the 'merry polka by some Austrian operetta composer . . . had become a funeral march dragging along at a grotesquely sluggish pace' (p. 348). While offering—like the antique shop—no genuine comprehension, this drastic alteration in tempo inspires some sense of connection; indeed, at one point he even believes he might have seen his mother, when he spots a woman who 'looks . . . just as [he] imagined the singer Agáta from [his] faint memories' (p. 351).

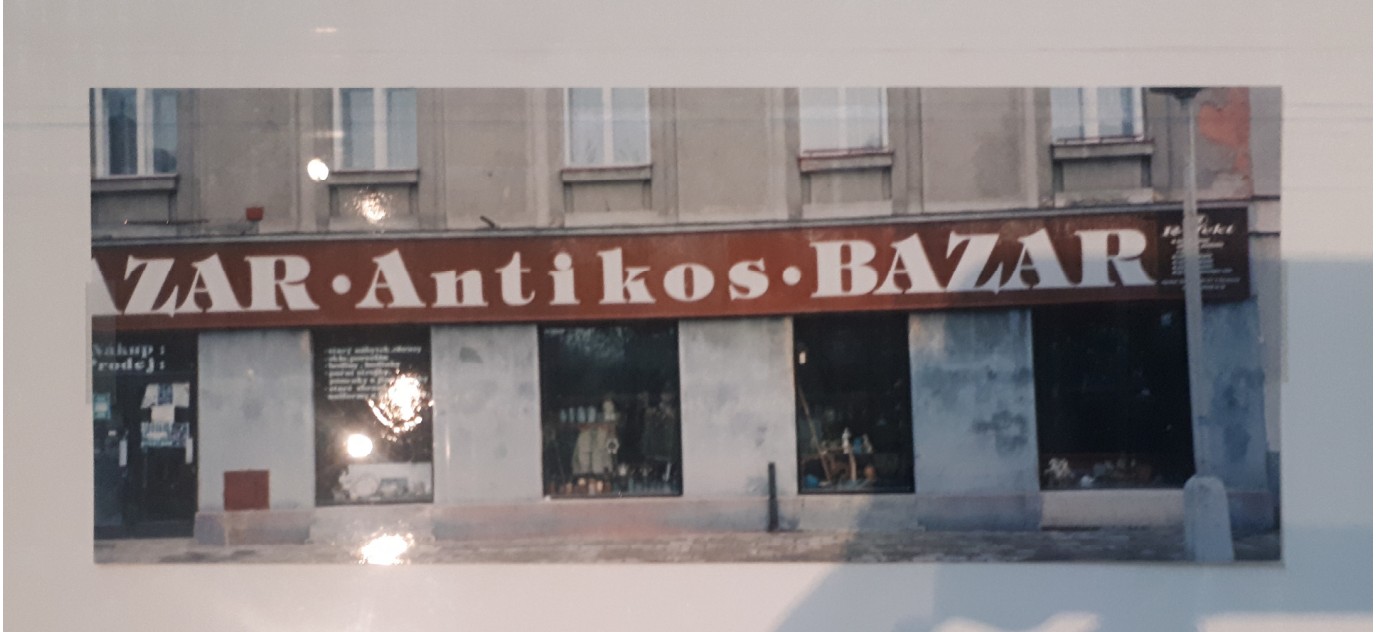

**Figure 1.** Sebald's original photograph of the Antikos Bazar in Terezín; author's own image, taken at the exhibition 'W.G. Sebald: Far Away, But From Where?' held at the Sainsbury Centre for Visual Arts, University of East Anglia, May-August 2019. The image is reproduced in black and white within the text (pp. 272–73).

In both of the above examples—and in stark contrast to the Ghetto Museum and the video in its full-speed form—we can observe a fluid relationship to time that recalls the

'stereometry' of interlocking spaces of the narrative and, concordantly, the fluid cartography necessary to its navigation. In Austerlitz's recollection—presumably from his childhood—of the word '*veverka*,' and his resistance to the Nazis' narrative temporality in the tightly controlled video of Terezín, a valuable (if unquantifiable and somewhat esoteric) sense of connection to his past emerges. Crucially, this is a sense that rests on fluidity, associative connection, and an impression of slowing down time, in contrast to the more rigid (if still valuable) information on display in the Ghetto Museum and the original form of the video. There is a striking sense here of a 'borderland,' to use Gil and Duarte's terms, between past and present—one which the protagonist can, it seems, freely traverse (Gil and Duarte 2011, p. 3). Through 'allegorical indirection' (Theisen 2006, p. 563) and a 'spherical system of association' (p. 569) Austerlitz paints a picture of his excavation of buried memory across a range of spaces. In turn, this impression is conveyed to the narrator and, hence, the reader, thereby adding additional 'slices of time' to the complex structure of spaces which make up the narrative. In following the charting of this layering-process, we are left with the distinct impression that Austerlitz can move at will between the pieces of the 'higher form of stereometry' that underpins the text (p. 261).

### 4. Conclusions: Crossing the Threshold

I would like to conclude here by introducing one final moment—specifically, a rare moment in which Sebald includes a reproduction of cartography, and a moment which can be considered a particularly neat encapsulation of the narrative method hitherto described. This is particularly true with respect to the notion that 'non–grid-like connections between people, places, times, and objects proliferate in *Austerlitz*,' and that the text functions, on some level, as a reaction to 'grid-like patterns' as a signifier of (rational) control (García-Moreno 2013, p. 367). It is perhaps unsurprising, however, that this moment can be considered paradigmatic, given that it occurs at the moment the eponymous protagonist experiences a breakdown, which leads him to begin a series of 'nocturnal wanderings through London, to escape . . . insomnia,' and which in turn lead to an encounter that reveals the true nature of his origins (p. 178). Typically for Sebald, these walking routes are furnished with a wealth of real-world geographical information:

> For over a year, I think, said Austerlitz, I would leave my house as darkness fell, walking on and on, down the Mile End Road and Bow Road to Stratford, right across Bethnal Green and Canonbury, through Holloway and Kentish Town and thus to Hampstead Heath, or else south over the river to Peckham and Dulwich or westward to Richmond Park. (p. 178)

As indicated by these locations, Austerlitz's nocturnal routes are enormous, and encapsulate much of London. It is during the course of one of these nightly perambulations—upon which he finds himself 'irresistibly drawn' to Liverpool Street Station with particular frequency and force—that the re-emergence of suppressed memories strikes him in the manner of a kind of traumatic revelation (p. 180).

Here, Sebald reproduces an historic map of Liverpool Street Station and Its surrounds, which depicts the station itself and a number of the streets thereabouts; this is the setting for the protagonist's involuntary reckoning with his own history. Arriving in the now-defunct 'Ladies Waiting-Room,' Austerlitz is assailed by a sense of shock which leaves him 'unable to move from the spot,' feeling like 'an actor who, upon making his entrance, has completely and irrevocably forgotten not only the lines he knew by heart but the very part he has so often played' (p. 189). Gradually after much more (sometimes, near-hallucinatory) confusion, Austerlitz begins to picture in his mind's eye at the station 'not only the minister and his wife'—referring to his Welsh foster parents—but also 'the boy they had come to meet': himself (p. 193). '[F]or the first time in as far back as I can remember,' he continues, 'I recollected myself as a small child, at the moment when I realized that it must have been to this same waiting-room I had come on my arrival to England over half a century ago' (p. 193). Within the text, it is this moment that represents Austerlitz's first true reckoning

with *memory* proper, as opposed to the 'spherical system of association' which has hitherto acted as a stand-in (Theisen 2006, p. 569).

Here, then, the wanderings that characterise Sebald's texts have led to memory, and to a moment of explicit revelation for Austerlitz. As ever, the preceding pages are replete with an array of transhistorical musings, which accompany the protagonist as he makes his nocturnal way across the streets of the capital. Particularly striking is a reference to Liverpool Street Station as akin to 'a kind of entrance to the underworld' (p. 180) prior to its renovation at the end of the 1980s, given its oppressively gloomy interior (and in fact, these renovation works coincide with the arrival of Austerlitz on his nightly walks, making the station a near-literal exemplar of a borderland between past and present). Likewise, discussion of the excavation of a nineteenth-century cemetery during these same works allows Austerlitz to describe, via the narrator, how the transformation of the station 'brought to light over four hundred skeletons underneath a taxi rank' (p. 184). Given what follows—the revelation of Austerlitz's own, hitherto-lost Holocaust-related past—the significance of these moments—couched as they are in language relating to the 'underworld,' 'excavation,' 'bringing to light,' 'eternal dusk,' and 'remains'—is quite clear; in their (allusive) inclusion, Sebald paints the area around Liverpool Street Station as one marked by death, and this is precisely the territory charted by the reproduced map.

In all of this, there are echoes of much of what has already been discussed above, here condensed into an intense sequence that represents the major point of departure for the narrative of *Austerlitz*—the moment of revelation, and the trigger for the protagonist's mnemonic tour of Europe in search of his origins. With regard to *Austerlitz* itself, this moment typifies the ghostly stereometry that characterises the text at large, providing as it does a further example of the interlocking 'slices of time' that are mapped out within. The 'territory' around Liverpool Street Station and the station itself represent 'not … borders but rather … borderlands' between past and present, as evidenced by the extensive descriptions provided by Sebald of historical events from the surrounding streets and his vivid memories of his own child-self disembarking from a *Kindertransport* train and encountering his adoptive parents for the first time (Gil and Duarte 2011, p. 3). This, really, is the key threshold here; as Austerlitz passes into the past—stepping from one space into another, lost space—he makes the move from 'allegorical indirection' and allusive association that have hitherto hinted toward his origins, and into the realm of direct memory (Theisen 2006, p. 563). As is the case elsewhere in the narrative, the attendant spatial description conveys a fluid cartography, given that the movement traced is not merely from space to space, but from *time* to *time*, too. Liverpool Street Station becomes the 'contact zone' *par excellence*, straddling as it does not only past and present spaces and events, but also the threshold between memory and oblivion (Gil and Duarte 2011, p. 3).

With respect to Sebald's wider oeuvre, too, there are aspects of this sequence that are paradigmatic of the author's narrative method. As we have seen, the nightly walks that led to the moment of revelation are related via specific (indeed, broadly mappable) geographical information. From the area surrounding the station, meanwhile, the protagonist embarks upon extended (and allusive) historical musings, including description of the station in its pre-renovation state, details relating to the nineteenth-century graveyard, and a variety of other topics. This is a feature of the narrative that is shared more widely throughout *Austerlitz*, although I have tended to focus here upon the notion of a 'stereometry of memory' that constitutes the building blocks of Sebald's last novel. In all, however, Sebald's works can be considered 'map-like' in character, charting as they do an array of specific locations, be they points upon a physical route, as we see clearly in *The Rings of Saturn* (Sebald 2002b; see Holden 2021) or more abstracted spaces, such as the 'borderlands' between past and present discussed in the present article. From these particular spaces, complex considerations of history and memory emerge, which tend to expand the spatial scope of the text to incorporate a wider web of (sometimes far-flung) places and spaces, not all of which are contemporaneous to the initial starting point. Given this fluid relation to space and time, Gil and Duarte's notion of 'fluid cartography' provides the most concise means by which to characterise the

map-like quality of these texts. Central to each narrative are the 'solitary travellers' (p. 7) who traverse Sebald's spaces, and who function as the 'carriers' of memory within each work (Erll 2011, p. 12). Throughout, the spectre of the Holocaust haunts both protagonists and spaces alike. In the words of the author, 'these subjects are constant company; their presence shades every inflection of every sentence one writes' (Jaggi 2001, ¶14).

**Funding:** This research received no external funding.

**Institutional Review Board Statement:** Not applicable.

**Informed Consent Statement:** Not applicable.

**Data Availability Statement:** No new data were created or analyzed in this study. Data sharing is not applicable to this article.

**Acknowledgments:** This work was completed as part of the author's PhD research, which was supported by The White Rose College of the Arts and Humanities. A version of this text appears as a chapter in the author's PhD thesis.

**Conflicts of Interest:** The author declares no conflict of interest.

## Notes

[1] This is a sense heightened by the somewhat-ghostly otherworldliness of the protagonist. We are told, for instance, that Austerlitz takes a number of photographs of mirrors in Antwerp. The narrator later finds that these particular images—rare photographs of the protagonist himself—have vanished from amidst the hundreds of pictures taken by Austerlitz (p. 7).

[2] Indeed, this sequence is expressed in a style very reminiscent of Kafka's *The Trial* (see Kafka 2015), and befitting of the term 'Kafkaesque': at the State Archives, Austerlitz finds himself overwhelmed by the environment, and suffers a panic attack (p. 208); his interlocutor is a seemingly ghostly 'woman of almost transparent appearance' (p. 206); finally, the trip to the archive also inspires 'fearful dreams' in which he finds himself '[climbing] up and down flights of steps, ringing hundreds of doorbells in vain' (p. 210). In all of this, we might well recall García-Moreno's assertion that *Austerlitz* details a certain aversion to the orderliness of the archive, and 'grid-like' control and regulation.

[3] This is despite its proximity to a major metropolis in the form of Frankfurt.

[4] There is too, we might observe, an assonant similarity in the words 'Vyrnwy' and 'Germany' when the two are spoken aloud.

[5] The latter of these question runs—in typical Sebaldian fashion—to a total of eighteen lines in its full iteration, and encompasses a wide array of items from the window display.

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
