# Peer review of "A Stereometry of Non-Memory: Mapping a Lost Past in W.G. Sebald’s Austerlitz"

_genealogy, doi:10.3390/genealogy7010015_

Round 1

Reviewer 1 Report

This is an interesting and thorough reading of Sebald's Austerlitz in light of theoretical engagement prompted by certain suggestive passages of the text. The notion of 'fluid cartography' is productive as a way of unlocking some of the text's intricacies. The article is well put together, consults existing scholarship, pushes debates forwards, and presents coherent conclusions clearly emanating from focused work on passages of the primary text. It is well written with all of the necessary apparatus in place, formatted according to expectations.

My main point for consideration and revision concerns the wording of the central claim and the implications that flow from that. The author argues that the protagonist's 'subjective experience of time is a consequence of the absence of memory experienced by the protagonist in relation to his origins as a Kindertransport survivor of the Holocaust.' As a result, spaces and buildings are found to compensate for this 'absence'. I don't think 'absence' is correct here. To my mind, the text is clear that Austerlitz's memories are repressed rather than absent. This might seem like a fairly small difference but it does carry implications: if memories are repressed rather than absent, then spaces and buildings act as prompts to their recovery rather than replacements for their loss. The article's position on this is not totally stable, with some slippage in the precise terminology used in the various constructions of the relationship. On page 2, association is said to act as a surrogate 'where there is no genuine recollection to be had'; conversely on page 3, there is discussion of 'disintegrated childhood memories' (that, presumably, might be reintegrated, and then of 'repressed Holocaust memory'; on page 4, there is a return to 'associative connection as a substitute for genuine memory', on page 8, the author doubles back again to a discussion of 'lost (or perhaps, repressed) memories' (my italics). The inconsistency here needs to be addressed.

Passages treated in later phases of the article make clear to me that memory is repressed rather than absent. On page 6, the author addresses Austerlitz's recovery of his ability to understand the Czech language. This linguistic ability, like the memories attached to it, are clearly repressed not absent. The discussion invokes Proust here - the analogy is a good one and bolsters the point.

I can suggest to ways to deal with this. The first is simply to revisit the expression, making key statements consistent. Obviously, I recommend that of the two options, the author opts for 'repressed' rather than 'absent' memories. The second is to think through and introduce some psychological models of repression, and maybe sublimation too, as responses to trauma. The novel seems to fit psychoanalysts' models of these mechanisms. This is a more thoroughgoing change with plenty of expansion of the article's range. The author and/or the editors may feel that this is beyond the scope of this piece and that the smaller change is all that is required.

Other than this issue, I enjoyed reading the article very much and feel that the journal's readership will too.

Author Response

I would like to thank the reviewer for taking the time to read this work, and for kindly offering such thoughtful and helpful responses; the suggested changes have helped to significantly clarify the focus of the article. I have adapted the text in order to address the inconsistency relating to absent vs. repressed memories. I have amended references to absent memory and changed them in order to signal repression, and I have removed most uses of the word ‘lost’ when referring to the memories or past of the protagonist. I have, however, sparingly retained (one or two) instances of the latter as they felt appropriate at some moments as a means of strongly signalling the absolute alienation (at least initially) of the protagonist from his childhood. I have elsewhere retained uses of the phrase ‘non-memory’/ ‘(non-)memory’ as a particular means of signalling the memory-surrogates that this article is concerned with, rather than as a means of suggesting that the protagonist lacks genuine memories or the capacity to remember; I hope that this does not detract from the clearer focus on repression that has been added or amended elsewhere.

Whilst I absolutely agree that the article does offer ample opportunity to discuss examples of repression/sublimation from within psychological and psychoanalytic literatures (particularly with the above clarifications now in place), I am afraid that I do not feel well-positioned to do justice to such a reading at present, particularly within the short time-scale requested by the journal for revisions. As such, I have opted to adhere to the first of the reviewer’s two suggestions for possible revisions.

Reviewer 2 Report

This is a coherently argued article that reflects and advances the present state of critical art in the field (ideas and narrative/poetic models of time and space in Sebald's novels, especially in Austerlitz) discussed. 

Author Response

I would like to thank the reviewer for taking the time to read the attached article, and for their kind comments.